# Most Neural Networks Are Almost Learnable

**Amit Daniely**
Hebrew University and Google
amit.daniely@mail.huji.ac.il

**Nathan Srebro**
TTI-Chicago
nati@ttic.edu

**Gal Vardi**
TTI-Chicago and Hebrew University
galvardi@ttic.edu

## Abstract

We present a PTAS for learning random constant-depth networks. We show that for any fixed $\epsilon > 0$ and depth $i$, there is a poly-time algorithm that for any distribution on $\sqrt{d} \cdot \mathbb{S}^{d-1}$ learns random Xavier networks of depth $i$, up to an additive error of $\epsilon$. The algorithm runs in time and sample complexity of $(\bar{d})^{\mathrm{poly}(\epsilon^{-1})}$, where $\bar{d}$ is the size of the network. For some cases of sigmoid and ReLU-like activations the bound can be improved to $(\bar{d})^{\mathrm{polylog}(\epsilon^{-1})}$, resulting in a quasi-poly-time algorithm for learning constant depth random networks.

## 1 Introduction

One of the greatest mysteries surrounding deep learning is the discrepancy between its phenomenal capabilities in practice and the fact that despite a great deal of research, polynomial-time algorithms for learning deep models are known only for very restrictive cases. Indeed, state of the art results are only capable of dealing with two-layer networks under assumptions on the input distribution and the network's weights. Furthermore, theoretical study shows that even with very naive architectures, learning neural networks is worst-case computationally intractable.

In this paper, we contrast the aforementioned theoretical state of affairs, and show that, perhaps surprisingly, even though constant-depth networks are completely out of reach from a worst-case perspective, *most* of them are not as hard as one would imagine. That is, they are *distribution-free learnable* in polynomial time up to any desired constant accuracy. This is the first polynomial-time approximation scheme (PTAS) for learning neural networks of depth greater than 2 (see the related work section for more details). Moreover, we show that the standard SGD algorithm on a ReLU network can be used as a PTAS for learning random networks. The question of whether learning random networks can be done efficiently was posed by Daniely et al. [15], and our work provides a positive result in that respect.

In a bit more detail, we consider constant-depth random networks obtained using the standard Xavier initialization scheme [22, 26], and any input distribution supported on the sphere $\sqrt{d} \cdot \mathbb{S}^{d-1}$. For Lipschitz activation functions, our algorithm runs in time $(\bar{d})^{\mathrm{poly}(\epsilon^{-1})}$, where $\bar{d}$ is the network's size including the $d$ input components, and $\epsilon$ is the desired accuracy. While this complexity is polynomial for constant $\epsilon$, we also consider the special cases of sigmoid and ReLU-like activations, where the bound can be improved to $(\bar{d})^{\mathrm{polylog}(\epsilon^{-1})}$.

The main technical idea in our work is that constant-depth random neural networks with Lipschitz activations can be approximated sufficiently well by low-degree polynomials. This result follows by analyzing the network obtained by replacing each activation function with its polynomial approximation using Hermite polynomials. It implies that efficient algorithms for learning polynomials can be

used for learning random neural networks, and specifically that we can use the SGD algorithm on ReLU networks for this task.

## 1.1 Results

In this work, we show that random fully-connected feedforward neural networks can be well-approximated by low-degree polynomials, which implies a PTAS for learning random networks. We start by defining the network architecture. We will denote by $\sigma : \mathbb{R} \to \mathbb{R}$ the activation function, and will assume that it is $L$-Lipschitz. To simplify the presentation, we will also assume that it is normalized in the sense that $\mathbb{E}_{X \sim \mathcal{N}(0,1)} \sigma^2(X) = 1$. Define $\epsilon_\sigma(n) = \min_{\deg(p)=n} \mathbb{E}_{X \sim \mathcal{N}(0,1)} (\sigma(X) - p(X))^2$, namely, the error when approximating $\sigma$ with a degree-$n$ polynomial, and note that $\lim_{n \to \infty} \epsilon_\sigma(n) = 0$. We will consider fully connected networks of depth $i$ and will use $d_0 = d$ to denote the input dimension and $d_1, \ldots, d_i$ to denote the number of neurons in each layer. Denote also $\bar{d} = \sum_{j=0}^{i} d_j$. Given weight matrices

$$\vec{W} = (W^1, \ldots, W^i) \in \mathbb{R}^{d_1 \times d_0} \times \ldots \times \mathbb{R}^{d_i \times d_{i-1}}$$

and $\mathbf{x} \in \mathbb{R}^{d_0}$ we define $\Psi^0_{\vec{W}}(\mathbf{x}) = \mathbf{x}$. Then for $1 \leq j \leq i$ we define recursively

$$\Phi^j_{\vec{W}}(\mathbf{x}) = W^j \Psi^{j-1}_{\vec{W}}(\mathbf{x}), \quad \Psi^j_{\vec{W}}(\mathbf{x}) = \sigma\left(\Phi^j_{\vec{W}}(\mathbf{x})\right)$$

We will consider random networks in which the weight matrices are random *Xavier matrices [22, 26]*. That is, each entry in $W^j$ is a centered Gaussian of variance $\frac{1}{d_{j-1}}$. This choice is motivated by the fact that it is a standard practice to initialize the network's weights with Xavier matrices, and furthermore, it ensures that the scale across the network is the same. That is, for any example $\mathbf{x}$ and a neuron $n$, the second moment of the output of $n$ (w.r.t. the choice of $\vec{W}$) is 1.

Our main result shows that $\Psi^i_{\vec{W}}$ can be approximated, up to any constant accuracy $\epsilon$, via constant degree polynomials (the constant will depend only on $\epsilon$, the depth $i$, and the activation $\sigma$). We will consider the input space $\tilde{\mathbb{S}}^{d-1} = \{\mathbf{x} \in \mathbb{R}^d : \|\mathbf{x}\| = 1\}$. Here, and throughout the paper, $\|\mathbf{x}\|$ stands for the *normalized* Euclidean norm $\|\mathbf{x}\| = \sqrt{\frac{1}{d} \sum_{i=1}^{d} x_i^2}$.

**Theorem 1.1.** *For every $i$ and $n$ such that $\epsilon_\sigma(n) \leq \frac{1}{2}$ there is a constant $D = D(n, i, \sigma)$ such that if $d_1, \ldots, d_{i-1} \geq D$ the following holds. For any weights $\vec{W}$, there is a degree $n^{i-1}$ polynomial $p_{\vec{W}}$ such that for any distribution $\mathcal{D}$ on $\tilde{\mathbb{S}}^{d-1}$*

$$\mathbb{E}_{\vec{W}} \mathbb{E}_{\mathbf{x} \sim \mathcal{D}} \left\| \Phi^i_{\vec{W}}(\mathbf{x}) - p_{\vec{W}}(\mathbf{x}) \right\| \leq 14 \cdot (L+1)^2 \cdot (\epsilon_\sigma(n))^{\frac{1}{2^{i-1}}} \leq \frac{14 \cdot (L+1)^3}{n^{\frac{1}{2^{i-1}}}} .$$

*Furthermore, the coefficients of $p_{\vec{W}}$ are bounded by $(2\bar{d})^{4n^{i-1}}$.*

Since constant degree polynomials are learnable in polynomial time, Theorem 1.1 implies a PTAS for learning random networks of constant depth. In fact, as shown in [9], constant degree polynomials with polynomial coefficients are efficiently learnable via SGD on ReLU networks starting from standard Xavier initialization. Thus, this PTAS can be standard SGD on neural networks. To be more specific, for any constant $\epsilon > 0$ there is an algorithm with $(\bar{d})^{O\left(\left(\frac{14(L+1)^3}{\epsilon}\right)^{(i-1)2^{i-1}}\right)}$ time and sample complexity that is guaranteed to return a hypothesis whose loss is at most $\epsilon$ in expectation. For some specific activations, such as the sigmoid $\sigma(x) = \mathrm{erf}(x) := \frac{2}{\sqrt{\pi}} \int_0^x e^{-\frac{t^2}{2}} dt$, or the ReLU-like activation $\sigma(x) = \int_0^x \mathrm{erf}(t) + 1 dt$ we have that $\epsilon_\sigma(n)$ approaches to 0 exponentially fast (see Lemma A.4 in the appendix). In this case, we get get a *quasi-polynomial* time and sample complexity of $(\bar{d})^{O\left(\left(\log\left(\frac{14(L+1)^3}{\epsilon}\right)\right)^{(i-1)}\right)}$.

**Corollary 1.2.** *For every constants $\epsilon, i$ and $\sigma$ there is a constant $D$, a univariate-polynomial $p$ and a polynomial-time algorithm $\mathcal{A}$ such that if $d_1, \ldots, d_{i-1} \geq D$ the following holds. For any distribution*

$\mathcal{D}$ on $\tilde{\mathbb{S}}^{d-1}$, if $h$ is the output of $\mathcal{A}$ upon seeing $p(d_0, \ldots, d_i)$ examples from $\mathcal{D}$, then[1]

$$\mathbb{E}_h \mathbb{E}_{\vec{W}} \mathbb{E}_{\mathbf{x} \sim \mathcal{D}} \left\| \Phi^i_{\vec{W}}(\mathbf{x}) - h(\mathbf{x}) \right\| \leq \epsilon \,.$$

*Furthermore, $\mathcal{A}$ can be taken to be SGD on a ReLU network starting from a Xavier initialization.*

## 1.2   Related Work

**Learning neural networks efficiently.**    Efficiently learning classes of neural networks has attracted much interest in recent years. Several works established polynomial-time algorithms for learning one-hidden-layer neural networks with certain input distributions (such as the Gaussian distribution) under the assumption that the weight matrix of the hidden layer is non-degenerate [27, 34, 19, 20, 5, 32, 4]. For example, Awasthi et al. [4] showed such a result for non-degenerate one-hidden-layer ReLU networks with bias terms under Gaussian inputs, and also concluded that one-hidden-layer networks can be learned efficiently under the smoothed-analysis framework. Efficient algorithms for learning one-hidden-layer ReLU networks with Gaussian inputs were also shown in Diakonikolas et al. [18], Diakonikolas and Kane [17]. These results do not require non-degenerate weight matrices, but they require that the output layer weights are all positive, as well as a sub-linear upper bound on the number of hidden neurons. Chen et al. [8] recently showed an efficient algorithm for learning one-hidden-layer ReLU networks with Gaussian inputs, under the assumption that the number of hidden neurons is a constant. Note that all of the aforementioned works consider only one-hidden-layer networks. Chen et al. [7] gave an algorithm for learning deeper ReLU networks, whose complexity is polynomial in the input dimension but exponential in the other parameters (such as the number of hidden units, depth, spectral norm of the weight matrices, and Lipschitz constant of the overall network). Finally, several works established algorithms for learning neural networks, whose complexity is exponential unless we impose strong assumptions on the norms of both the inputs and the weights [23, 30, 33, 24].

**Hardness of learning neural networks.**    As we discussed in the previous paragraph, efficient algorithms for learning ReLU networks are known only for depth-2 networks and under certain assumptions on both the network weights and the input distribution. The limited progress in learning ReLU networks can be partially understood by an abundance of hardness results.

Learning neural networks without any assumptions on the input distribution or the weights is known to be hard (under cryptographic and average-case hardness assumptions) already for depth-2 ReLU networks [28, 3, 11]. For depth-3 networks, hardness results were obtained already when the input distribution is Gaussian [13, 6]. All of the aforementioned hardness results are for improper learning, namely, they do not impose any restrictions on the learning algorithm or on the hypothesis that it returns. For *statistical query (SQ)* algorithms, unconditional superpolynomial lower bounds were obtained for learning depth-3 networks with Gaussian inputs [6], and superpolynomial lower bounds for *Correlational SQ (CSQ)* algorithms were obtained already for learning depth-2 networks with Gaussian inputs [25, 18].

The above negative results suggest that assumptions on the input distribution may not suffice for obtaining efficient learning algorithms. Since in one-hidden-layer networks efficient algorithms exist when imposing assumptions on both the input distribution and the weights, a natural question is whether this approach might also work for deeper networks. Recently, Daniely et al. [15] gave a hardness result for improperly learning depth-3 ReLU networks under the Gaussian distribution even when the weight matrices are non-degenerate. This result suggests that learning networks of depth larger than 2 might require new approaches and new assumptions. Moreover, [15] showed hardness of learning depth-3 networks under the Gaussian distribution even when a small random perturbation is added to the network's parameters, namely, they proved hardness in the smoothed-analysis framework. While adding a small random perturbation to the parameters does not seem to make the problem computationally easier, they posed the question of whether learning random networks, which roughly correspond to adding a large random perturbation, can be done efficiently. The current work gives a positive result in that respect.

Daniely and Vardi [12] studied whether there exist some "natural" properties of the network's weights that may suffice to allow efficient distribution-free learning, where a "natural" property is any property

---

[1]The leftmost expectation denoted $\mathbb{E}_h$ is over the examples provided to $\mathcal{A}$, as well as the internal randomness of $\mathcal{A}$.

that holds w.h.p. in random networks. More precisely, they considered a setting where the target network is random, an adversary chooses some input distribution (that may depend on the target network), and the learning algorithm needs to learn the random target network under this input distribution. They gave a hardness result for improper learning (within constant accuracy) in this setting. Thus, they showed that learning random networks is hard when the input distribution may depend on the random network. Note that in the current work, we give a positive result in a setting where we first fix an input distribution and then draw a random network. Finally, learning deep random networks was studied in Das et al. [16], Agarwal et al. [1], where the authors showed hardness of learning networks of depth $\omega(\log(d))$ in the SQ model.

## 2 Proof of Theorem 1.1

### 2.1 Notation

We recall that for vectors $\mathbf{x} \in \mathbb{R}^d$ we use the *normalized* Euclidean norm $\|\mathbf{x}\| = \sqrt{\frac{\sum_{i=1}^d x_i^2}{d}}$ and take the unit sphere $\tilde{\mathbb{S}}^{d-1} = \{\mathbf{x} \in \mathbb{R}^d : \|\mathbf{x}\| = 1\}$ w.r.t. this norm as our instance space. Inner products will also be normalized: for $\mathbf{x}, \mathbf{y} \in \mathbb{R}^d$ we denote $\langle \mathbf{x}, \mathbf{y} \rangle = \frac{\sum_{i=1}^d x_i y_i}{d}$. For $\mathbf{x} \in \mathbb{R}^d$ and a closed set $A \subset \mathbb{R}^d$ we denote $d(\mathbf{x}, A) := \min_{\mathbf{x}' \in A} \|\mathbf{x} - \mathbf{x}'\|$. Unless otherwise specified, a random scalar is assumed to be a standard normal, a random vector in $\mathbb{R}^d$ is assumed to be a centered Gaussian vector with covariance matrix $\frac{1}{d}I$, and a random matrix is assumed to be a Xavier matrix. For $f : \mathbb{R} \to \mathbb{R}$, we denote $\|f\|^2 = \mathbb{E}_X f^2(X)$. We denote the Kronecker delta by $\delta_{ij}$, i.e. $\delta_{ij} = 1$ if $i = j$ and $0$ otherwise.

### 2.2 Some Preliminaries

We will use the Hermite Polynomials [29] which are defined via the following recursion formula.

$$h_{n+1}(x) = \frac{x}{\sqrt{n+1}} h_n(x) - \sqrt{\frac{n}{n+1}} h_{n-1}(x), \;\; h_0(x) = 1, \;\; h_1(x) = x \tag{1}$$

The Hermite polynomials are the sequence of normalized orthogonal polynomials w.r.t. the standard Gaussian measure. That is, it holds that

$$\mathbb{E}_X h_i(X) h_j(X) = \delta_{ij}$$

More generally, if $(X, Y)$ is a Gaussian vector with covariance matrix $\begin{pmatrix} 1 & \rho \\ \rho & 1 \end{pmatrix}$ then

$$\mathbb{E}_{X,Y} h_i(X) h_j(Y) = \delta_{ij} \rho^i \tag{2}$$

We will use the fact that

$$h_n' = \sqrt{n} h_{n-1} \tag{3}$$

and that for even $n$

$$\mathbb{E}_X X^n = (n-1)!! \tag{4}$$

Let $\sigma = \sum_{i=0}^{\infty} a_i h_i$ be the representation of the activation function $\sigma$ in the basis of the Hermite polynomials. We will also use the *dual activation* $\hat{\sigma}(\rho) = \sum_{i=0}^{\infty} a_i^2 \rho^i$ as defined in [14]. We note that $\hat{\sigma}$ is defined in $[-1, 1]$ and satisfies $\hat{\sigma}(1) = \|\sigma\|^2 = 1$.

### 2.3 Defining a Shadow Network

In order to approximate $\Psi_{\vec{W}}^i$ via a polynomial, we will use a "shadow network" that is obtained by replacing the activation $\sigma$ with a polynomial approximation of it. We will show that for random networks we can approximate each activation sufficiently well with low-degree Hermite polynomials. Recall that $\sigma = \sum_{i=0}^{\infty} a_i h_i$ is the representation of $\sigma$ in the basis of the Hermite polynomials. Define $\sigma_n = \frac{1}{\sqrt{\sum_{i=0}^n a_i^2}} \sum_{i=0}^n a_i h_i$. We have $\epsilon_\sigma(n) = \sum_{i=n+1}^{\infty} a_i^2$ and hence $\sigma_n = \frac{1}{\sqrt{1 - \epsilon_\sigma(n)}} \sum_{i=0}^n a_i h_i$.

We next define a shadow network. For $\mathbf{x} \in \mathbb{R}^d$ we let $\Psi_{\vec{W}}^{0,n}(\mathbf{x}) = \mathbf{x}$. For $1 \le j \le i$ we define recursively

$$\Phi_{\vec{W}}^{j,n}(\mathbf{x}) = W^j \Psi_{\vec{W}}^{j-1,n}(\mathbf{x}), \quad \Psi_{\vec{W}}^{j,n}(\mathbf{x}) = \sigma_n\left(\Phi_{\vec{W}}^{j,n}(\mathbf{x})\right)$$

for $1 \le j \le i-1$ and $\Psi_{\vec{W}}^{i,n}(\mathbf{x}) = W^i \Psi_{\vec{W}}^{i-1,n}(\mathbf{x})$. We will prove the following theorem, which implies Theorem 1.1.

**Theorem 2.1.** *Fix $i$ and let $n$ be large enough so that $\epsilon_\sigma(n) \le \frac{1}{2}$. There is a constant $D = D(n,i,\sigma)$ such that if $d_1, \dots, d_{i-2} \ge D$ then for any $\mathbf{x} \in \tilde{\mathbb{S}}^{d-1}$,*

$$\mathbb{E}_{\vec{W}} \left\| \Phi_{\vec{W}}^i(\mathbf{x}) - \Phi_{\vec{W}}^{i,n}(\mathbf{x}) \right\| \le 13 \cdot (L+1)^2 \cdot (\epsilon_\sigma(n))^{\frac{1}{2^{i-1}}}$$

Since $\epsilon_\sigma(n)$ is the error in the approximation of a single activation $\sigma$ with a degree-$n$ polynomial, it is natural to expect that the above bound will depend on $\epsilon_\sigma(n)$. To see why Theorem 2.1 (together with Lemma A.3 which bounds $\epsilon_\sigma(n)$) implies Theorem 1.1, note that $\Phi_{\vec{W}}^{i,n}(\mathbf{x})$ is a polynomial of degree $n^{i-1}$. This implies Theorem 1.1, except the requirement that the coefficients of the polynomial are polynomially bounded. To deal with this, define

$$\tilde{\Phi}_{\vec{W}}^{i,n}(\mathbf{x}) = \begin{cases} \Phi_{\vec{W}}^{i,n}(\mathbf{x}) & \text{if all entries in } \vec{W} \text{ are at most } \sum_{j=0}^{i} d_j \\ 0 & \text{otherwise} \end{cases}$$

As we show next $\lim_{\min(d_1,\dots,d_{i-1})\to\infty} \mathbb{E}_{\vec{W}} \left\| \Phi_{\vec{W}}^{i,n}(\mathbf{x}) - \tilde{\Phi}_{\vec{W}}^{i,n}(\mathbf{x}) \right\| = 0$. Hence, in the theorem we can replace $\Phi^{i,n}$ by $\tilde{\Phi}^{i,n}$ which has polynomially bounded coefficients. See Appendix A.3 and A.4 for the proofs.

**Lemma 2.2.** *For every $\epsilon$ and $n$ there is a constant $D$ such that if $d_1, \dots, d_{i-1} \ge D$ then for any $\mathbf{x} \in \tilde{\mathbb{S}}^{d-1}$, $\mathbb{E}_{\vec{W}} \left\| \Phi_{\vec{W}}^{i,n}(\mathbf{x}) - \tilde{\Phi}_{\vec{W}}^{i,n}(\mathbf{x}) \right\| < \epsilon$.*

**Lemma 2.3.** *$\tilde{\Phi}_{\vec{W}}^{i,n}$ computes a polynomial whose sum of coefficients is at most $(2\bar{d})^{4n^{i-1}}$.*

## 2.4 Proof of Theorem 2.1 for depth-two networks

We will first prove Theorem 2.1 for depth-2 networks (i.e. for $i = 2$). We will prove Lemma 2.5 below which implies that for every $\epsilon$ there is $n$ such that for any $\mathbf{x} \in \tilde{\mathbb{S}}^{d-1}$, $\mathbb{E}_{\vec{W}} \left\| \Psi_{\vec{W}}^{1,n}(\mathbf{x}) - \Psi_{\vec{W}}^1(\mathbf{x}) \right\| \le \epsilon$. We will then prove Lemma 2.6, that together with Lemma 2.5 will show that $\mathbb{E}_{\vec{W}} \left\| \Phi_{\vec{W}}^{2,n}(\mathbf{x}) - \Phi_{\vec{W}}^2(\mathbf{x}) \right\| \le \epsilon$, thus proving Theorem 2.1 for $i = 2$. We will start however with the following lemma that will be useful throughout (see Appendix A.5 for the proof).

**Lemma 2.4.** *Fix $f, g : \mathbb{R} \to \mathbb{R}$, $\mathbf{x}, \mathbf{y} \in \mathbb{R}^{d_1}$ and a Xavier matrix $W \in \mathbb{R}^{d_2 \times d_1}$. Let $(X, Y)$ be a centered Gaussian vector with covariance matrix $\begin{pmatrix} \|\mathbf{x}\|^2 & \langle \mathbf{x}, \mathbf{y} \rangle \\ \langle \mathbf{x}, \mathbf{y} \rangle & \|\mathbf{y}\|^2 \end{pmatrix}$. Then*

$$\mathbb{E}_W \| f(W\mathbf{x}) - g(W\mathbf{y}) \| \le \sqrt{\mathbb{E}_W \| f(W\mathbf{x}) - g(W\mathbf{y}) \|^2} = \sqrt{\mathbb{E}_{X,Y} (f(X) - g(Y))^2}$$

**Lemma 2.5.** *Fix $\mathbf{x} \in \tilde{\mathbb{S}}^{d_1-1}$. Let $W \in \mathbb{R}^{d_2 \times d_1}$ be a Xavier matrix. Then*

$$\mathbb{E}_W \| \sigma(W\mathbf{x}) - \sigma_n(W\mathbf{x}) \| \le \sqrt{2\epsilon_\sigma(n)}$$

*Proof.* By Lemma 2.4 we have

$$\mathbb{E}_W \| \sigma(W\mathbf{x}) - \sigma_n(W\mathbf{x}) \| \le \sqrt{\mathbb{E}_W \| \sigma(W\mathbf{x}) - \sigma_n(W\mathbf{x}) \|^2} = \sqrt{\mathbb{E}_X (\sigma(X) - \sigma_n(X))^2} \,.$$

Now, the above equals to

$$\sqrt{\sum_{i=0}^{n}\left(1-\frac{1}{\sqrt{1-\epsilon_\sigma(n)}}\right)^2 a_i^2 + \sum_{i=n+1}^{\infty} a_i^2} = \sqrt{(1-\epsilon_\sigma(n))\left(1-\frac{1}{\sqrt{1-\epsilon_\sigma(n)}}\right)^2 + \epsilon_\sigma(n)}$$

$$= \sqrt{(1-\epsilon_\sigma(n))\left(\frac{\sqrt{1-\epsilon_\sigma(n)}-1}{\sqrt{1-\epsilon_\sigma(n)}}\right)^2 + \epsilon_\sigma(n)}$$

$$= \sqrt{2 - \epsilon_\sigma(n) - 2\sqrt{1-\epsilon_\sigma(n)} + \epsilon_\sigma(n)}$$

$$= \sqrt{2(1-\sqrt{1-\epsilon_\sigma(n)})}$$

$$\leq \sqrt{2(1-\sqrt{1-\epsilon_\sigma(n)})(1+\sqrt{1-\epsilon_\sigma(n)})}$$

$$= \sqrt{2\epsilon_\sigma(n)}$$

$\square$

Lemma 2.5 implies that $\mathbb{E}_{\vec{W}}\left\|\Psi_{\vec{W}}^{1,n}(\mathbf{x}) - \Psi_{\vec{W}}^{1}(\mathbf{x})\right\| \leq \sqrt{2\epsilon_\sigma(n)}$. Thus, given $\epsilon > 0$, for sufficiently large $n$, $\mathbb{E}_{\vec{W}}\left\|\Psi_{\vec{W}}^{1,n}(\mathbf{x}) - \Psi_{\vec{W}}^{1}(\mathbf{x})\right\| \leq \epsilon$. The following lemma therefore implies that $\mathbb{E}_{\vec{W}}\left\|\Phi_{\vec{W}}^{2,n}(\mathbf{x}) - \Phi_{\vec{W}}^{2}(\mathbf{x})\right\| \leq \sqrt{2\epsilon_\sigma(n)}$ and thus implies Theorem 2.1 for depth two networks.

**Lemma 2.6.** *For any* $\mathbf{x} \in \tilde{\mathbb{S}}^{d-1}$

$$\mathbb{E}_{W^i}\left\|\Phi_{\vec{W}}^{i,n}(\mathbf{x}) - \Phi_{\vec{W}}^{i}(\mathbf{x})\right\| \leq \left\|\Psi_{\vec{W}}^{i-1,n}(\mathbf{x}) - \Psi_{\vec{W}}^{i-1}(\mathbf{x})\right\|$$

*Proof.* By Lemma 2.4 we have

$$\mathbb{E}_{W^i}\left\|\Phi_{\vec{W}}^{i,n}(\mathbf{x}) - \Phi_{\vec{W}}^{i}(\mathbf{x})\right\| = \mathbb{E}_{W^i}\left\|W^i\left(\Psi_{\vec{W}}^{i-1,n}(\mathbf{x}) - \Psi_{\vec{W}}^{i-1}(\mathbf{x})\right)\right\|$$

$$\leq \sqrt{\mathbb{E}_{X\sim\mathcal{N}\left(0,\left\|\Psi_{\vec{W}}^{i-1,n}(\mathbf{x})-\Psi_{\vec{W}}^{i-1}(\mathbf{x})\right\|^2\right)} X^2}$$

$$= \left\|\Psi_{\vec{W}}^{i-1,n}(\mathbf{x}) - \Psi_{\vec{W}}^{i-1}(\mathbf{x})\right\|$$

$\square$

## 2.5 Proof of Theorem 2.1 for General Networks

For $\mathbf{x} \in \tilde{\mathbb{R}}^{d_{i-1}}$ we denote $\Psi_{W^i}(\mathbf{x}) = \sigma(W^i\mathbf{x})$ and $\Psi_{W^i}^n(\mathbf{x}) = \sigma_n(W^i\mathbf{x})$. Lemma 2.5 can be roughly phrased as

$$(\mathbf{x} = \mathbf{x}') \text{ and } (\|\mathbf{x}\| = 1) \Rightarrow \Psi_{W^i}(\mathbf{x}) \approx \Psi_{W^i}^n(\mathbf{x}')$$

In order to prove Theorem 2.1 for general networks we will extend it by replacing the strict equality conditions with softer ones. That is, we will show that

$$(\mathbf{x} \approx \mathbf{x}') \text{ and } (\|\mathbf{x}\| \approx 1) \text{ and } (\|\mathbf{x}'\| \approx 1) \Rightarrow \Psi_{W^i}(\mathbf{x}) \approx \Psi_{W^i}^n(\mathbf{x}') \tag{5}$$

This will be enough to prove Theorem 2.1 for general networks. Indeed, the conditions $\|\mathbf{x}\| \approx 1$ and $\|\mathbf{x}'\| \approx 1$ are valid w.h.p. via a simple probabilistic argument. Thus, Eq. (5) implies that

$$\mathbf{x} \approx \mathbf{x}' \Rightarrow \Psi_{W^i}(\mathbf{x}) \approx \Psi_{W^i}^n(\mathbf{x}') \tag{6}$$

Now, for $\mathbf{x} \in \tilde{\mathbb{S}}^{d-1}$ Eq. (6) implies that $\Psi_{W^1}(\mathbf{x}) \approx \Psi_{W^1}^n(\mathbf{x}')$. Using Eq. (6) again we get that $\Psi_{W^2} \circ \Psi_{W^1}(\mathbf{x}) \approx \Psi_{W^2}^n \circ \Psi_{W^1}^n(\mathbf{x}')$. Using it $i-3$ more times we we get that $\Psi_{W^{i-1}} \circ \cdots \circ \Psi_{W^1}(\mathbf{x}) \approx \Psi_{W^{i-1}}^n \circ \Psi_{W^1}^n(\mathbf{x}')$, or in other words that $\Psi_{\vec{W}}^{i-1}(\mathbf{x}) \approx \Psi_{\vec{W}}^{i-1,n}(\mathbf{x})$. As we will show "$\approx$" stands for a sufficiently strong approximation, which guarantees that $\mathbb{E}_{\vec{W}}\|\Psi_{\vec{W}}^{i-1}(\mathbf{x}) - \Psi_{\vec{W}}^{i-1,n}(\mathbf{x})\| \leq \epsilon$, and hence Lemma 2.6 implies Theorem 2.1.

To prove Eq. (5) we first prove Lemma 2.7 which softens the requirement that $\mathbf{x} = \mathbf{x}'$. That is, it shows that

$$(\mathbf{x} \approx \mathbf{x}') \text{ and } (\|\mathbf{x}\| = \|\mathbf{x}'\| = 1) \Rightarrow \Psi_{W^i}(\mathbf{x}) \approx \Psi_{W^i}^n(\mathbf{x}')$$

The second condition which requires that $\|\mathbf{x}\| = \|\mathbf{x}'\| = 1$ is softened via Lemmas 2.8 and 2.9. Lemma 2.10 then wraps the two softenings together, and shows that Eq. (5) is valid. Finally, in section 2.5.1 we use Lemma 2.10 to prove Theorem 2.1.

**Lemma 2.7.** *Fix* $\mathbf{x}, \mathbf{x} + \mathbf{v} \in \tilde{\mathbb{S}}^{d_1 - 1}$ *with* $\|\mathbf{v}\| \leq \epsilon$. *Let* $W \in \mathbb{R}^{d_2 \times d_1}$ *be a Xavier matrix. Then*

$$\underset{W}{\mathbb{E}} \|\sigma(W\mathbf{x}) - \sigma_n(W(\mathbf{x} + \mathbf{v}))\| \leq \sqrt{2\epsilon_\sigma(n)} + \sqrt{\frac{2L^2}{1 - \epsilon_\sigma(n)}} \epsilon$$

*Proof.* We have

$$\|\sigma(W\mathbf{x}) - \sigma_n(W(\mathbf{x} + \mathbf{v}))\| \leq \|\sigma(W\mathbf{x}) - \sigma_n(W\mathbf{x})\| + \|\sigma_n(W\mathbf{x}) - \sigma_n(W(\mathbf{x} + \mathbf{v}))\|$$

By Lemma 2.5 we have $\mathbb{E}_W \|\sigma(W\mathbf{x}) - \sigma_n(W\mathbf{x})\| \leq \sqrt{2\epsilon_\sigma(n)}$. It remains to bound $\mathbb{E}_W \|\sigma_n(W\mathbf{x}) - \sigma_n(W(\mathbf{x} + \mathbf{v}))\|$. By Lemma 2.4 We have

$$\underset{W}{\mathbb{E}} \|\sigma_n(W\mathbf{x}) - \sigma_n(W(\mathbf{x} + \mathbf{v}))\| \leq \sqrt{\underset{X,Y}{\mathbb{E}} (\sigma_n(X) - \sigma_n(Y))^2}$$

where $(X, Y)$ is a centered Gaussian vector with correlation matrix $\begin{pmatrix} 1 & \rho \\ \rho & 1 \end{pmatrix}$ for $\rho = \langle \mathbf{x}, \mathbf{x} + \mathbf{v} \rangle \geq 1 - \epsilon$. Finally, we have

$$
\begin{aligned}
\underset{X,Y}{\mathbb{E}} (\sigma_n(X) - \sigma_n(Y))^2 &= \frac{1}{1 - \epsilon_\sigma(n)} \underset{X,Y}{\mathbb{E}} \left( \sum_{i=0}^n a_i (h_i(X) - h_i(Y)) \right)^2 \\
&= \frac{1}{1 - \epsilon_\sigma(n)} \sum_{i=0}^n \sum_{j=0}^n a_i a_j \underset{X,Y}{\mathbb{E}} (h_i(X) - h_i(Y))(h_j(X) - h_j(Y)) \\
&\overset{Eq.\ (2)}{=} \frac{1}{1 - \epsilon_\sigma(n)} \sum_{i=0}^n a_i^2 (2 - 2\rho^i) \\
&\leq \frac{2}{1 - \epsilon_\sigma(n)} (\hat{\sigma}(1) - \hat{\sigma}(\rho))
\end{aligned}
$$

In Lemma A.1 we show that $\hat{\sigma}$ is $L^2$-Lipschitz. Hence the above is at most $\frac{2L^2}{1 - \epsilon_\sigma(n)} \epsilon$. $\qquad\square$

We next give a lemma that allows us to "almost jointly project" a pair of points $\mathbf{x}_1, \mathbf{x}_2 \in \mathbb{R}^d$ on a closed set $A \subset \mathbb{R}^d$, without expanding the distance too much. See Appendix A.6 for the proof.

**Lemma 2.8.** *Let* $A \subset \mathbb{R}^d$ *a closed set and fix* $\mathbf{x}_1, \mathbf{x}_2 \in \mathbb{R}^d$. *There are* $\tilde{\mathbf{x}}_1, \tilde{\mathbf{x}}_2 \in A$ *such that*

$$\|\mathbf{x}_1 - \tilde{\mathbf{x}}_1\| \leq 2d(\mathbf{x}_1, A), \quad \|\mathbf{x}_2 - \tilde{\mathbf{x}}_2\| \leq 2d(\mathbf{x}_2, A) \quad and \quad \|\tilde{\mathbf{x}}_1 - \tilde{\mathbf{x}}_2\| \leq 3\|\mathbf{x}_1 - \mathbf{x}_2\|$$

**Lemma 2.9.** *Let* $\mathbf{x}, \mathbf{x} + \mathbf{v} \in \mathbb{R}^{d_1}$ *be vectors such that* $\|\mathbf{x}\| = 1$ *and* $\|\mathbf{v}\| \leq \epsilon \leq 1$. *Let* $W \in \mathbb{R}^{d_2 \times d_1}$ *be a Xavier matrix. Then*

$$\underset{W}{\mathbb{E}} \|\sigma(W\mathbf{x}) - \sigma(W(\mathbf{x} + \mathbf{v}))\| \leq L\epsilon$$

*and*

$$\underset{W}{\mathbb{E}} \|\sigma_n(W\mathbf{x}) - \sigma_n(W(\mathbf{x} + \mathbf{v}))\| \leq 2^{2n+1} (9(4n - 1)!!)^{1/4} \epsilon =: \lambda(n)\epsilon$$

*Proof.* Fix a centered Gaussain vector $(X, Y)$ with covariance matrix $\begin{pmatrix} 1 & \langle \mathbf{x} + \mathbf{v}, \mathbf{x} \rangle \\ \langle \mathbf{x} + \mathbf{v}, \mathbf{x} \rangle & \|\mathbf{x} + \mathbf{v}\|^2 \end{pmatrix}$. Let $Z = Y - X$. Note that $\mathrm{Var}(Z) \leq \epsilon^2$. By Lemma 2.4 we have

$$\underset{W}{\mathbb{E}} \|\sigma(W\mathbf{x}) - \sigma(W(\mathbf{x} + \mathbf{v}))\| \leq \sqrt{\mathbb{E}(\sigma(X) - \sigma(X + Z))^2} \leq \sqrt{L^2 \mathbb{E} Z^2} \leq L\epsilon$$

We now prove the second part. In Lemma A.2, we show that $|h_i(x) - h_i(x + y)| \leq 2^i \max(|x|, |x + y|, 1)^i |y|$. Therefore,

$$
\begin{aligned}
|\sigma_n(x) - \sigma_n(x + y)| &\leq \sum_{i=0}^{n} \frac{|a_i|}{\sqrt{1 - \epsilon_\sigma(n)}} |h_i(x) - h_i(x + y)| \\
&\leq |y| \sum_{i=0}^{n} 2^i \max(|x|^i, |x + y|^i, 1) \\
&\leq |y| 2^{n+1} \max(|x|^n, |x + y|^n, 1)
\end{aligned}
$$

Hence,

$$
\begin{aligned}
\mathbb{E}(\sigma_n(X) - \sigma_n(X + Z))^2 &\leq 2^{2n+2} \mathbb{E} Z^2 \max(|X|^n, |X + Z|^n, 1)^2 \\
&\leq 2^{2n+2} \sqrt{\mathbb{E} Z^4} \sqrt{\mathbb{E} \max(|X|^{4n}, |X + Z|^{4n}, 1)} \\
&\leq 2^{2n+2} \sqrt{\mathbb{E} Z^4} \sqrt{\mathbb{E} [|X|^{4n} + |X + Z|^{4n} + 1]} \\
&\stackrel{Eq. (4)}{=} 2^{2n+2} \sqrt{3\|\mathbf{v}\|^4} \sqrt{1 + (4n - 1)!!(\|\mathbf{x} + \mathbf{v}\|^{4n} + \|\mathbf{x}\|^{4n})} \\
&\leq 2^{2n+2} \sqrt{3\epsilon^4} \sqrt{3(1 + \epsilon)^{4n}(4n - 1)!!} \\
&\leq 2^{2n+2} \sqrt{3\epsilon^4} \sqrt{3 \cdot 2^{4n} \cdot (4n - 1)!!}
\end{aligned}
$$

Lemma 2.4 now implies that

$$
\mathbb{E}_W \|\sigma_n(W\mathbf{x}) - \sigma_n(W(\mathbf{x} + \mathbf{v}))\| \leq 2^{2n+1} (9(4n - 1)!!)^{1/4} \epsilon
$$

$\square$

**Lemma 2.10.** *Let $\mathbf{x}, \mathbf{x} + \mathbf{v} \in \mathbb{R}^{d_1}$ be vectors such that $\|\mathbf{v}\| \leq \epsilon$, $|\|\mathbf{x}\| - 1| \leq \delta \leq 1/2$ and $|\|\mathbf{x} + \mathbf{v}\| - 1| \leq \delta$. Let $W \in \mathbb{R}^{d_2 \times d_1}$ be a Xavier matrix. Then*

$$
\mathbb{E}_W \|\sigma(W\mathbf{x}) - \sigma_n(W(\mathbf{x} + \mathbf{v}))\| \leq 2L\delta + \sqrt{2\epsilon_\sigma(n)} + \sqrt{\frac{6L^2}{1 - \epsilon_\sigma(n)} \epsilon + 2\lambda(n)\delta}
$$

*Proof.* By Lemma 2.8 there are vectors $\mathbf{x}', \mathbf{v}'$ such that $\|\mathbf{x}'\| = \|\mathbf{x}' + \mathbf{v}'\| = 1$ and

$$
\|\mathbf{x} - \mathbf{x}'\| \leq 2\delta, \ \|\mathbf{x} + \mathbf{v} - \mathbf{x}' - \mathbf{v}'\| \leq 2\delta, \ \text{and} \ \|\mathbf{v}'\| \leq 3\|\mathbf{v}\|
$$

Now, we have, by Lemmas 2.7 and 2.9,

$$
\begin{aligned}
\mathbb{E}_W \|\sigma(W\mathbf{x}) - \sigma_n(W(\mathbf{x} + \mathbf{v}))\| &\leq \mathbb{E}_W \|\sigma(W\mathbf{x}) - \sigma(W\mathbf{x}')\| + \mathbb{E}_W \|\sigma(W\mathbf{x}') - \sigma_n(W(\mathbf{x}' + \mathbf{v}'))\| \\
&\quad + \mathbb{E}_W \|\sigma_n(W(\mathbf{x}' + \mathbf{v}')) - \sigma_n(W(\mathbf{x} + \mathbf{v}))\| \\
&\leq 2L\delta + \sqrt{2\epsilon_\sigma(n)} + \sqrt{\frac{6L^2}{1 - \epsilon_\sigma(n)} \epsilon + 2\lambda(n)\delta}
\end{aligned}
$$

$\square$

### 2.5.1 Concluding the proof of Theorem 2.1

Define

$$
\Psi_{\vec{W}}^{i}(\mathbf{x}, \delta) = \begin{cases} 0 & |1 - \|\Psi_{\vec{W}}^{j}(\mathbf{x})\|\| > \delta \text{ or } |1 - \|\Psi_{\vec{W}}^{j,n}(\mathbf{x})\|\| > \delta \text{ for some } j < i \\ \Psi_{\vec{W}}^{i}(\mathbf{x}) & \text{otherwise} \end{cases}
$$

and

$$
\Psi_{\vec{W}}^{i,n}(\mathbf{x}, \delta) = \begin{cases} 0 & |1 - \|\Psi_{\vec{W}}^{j}(\mathbf{x})\|\| > \delta \text{ or } |1 - \|\Psi_{\vec{W}}^{j,n}(\mathbf{x})\|\| > \delta \text{ for some } j < i \\ \Psi_{\vec{W}}^{i,n}(\mathbf{x}) & \text{otherwise} \end{cases}
$$

We have

$$\mathop{\mathbb{E}}_{\vec{W}}\left\|\Psi^i_{\vec{W}}(\mathbf{x}) - \Psi^{i,n}_{\vec{W}}(\mathbf{x})\right\| \le \mathop{\mathbb{E}}_{\vec{W}}\left\|\Psi^i_{\vec{W}}(\mathbf{x}) - \Psi^i_{\vec{W}}(\mathbf{x},\delta)\right\| + \mathop{\mathbb{E}}_{\vec{W}}\left\|\Psi^i_{\vec{W}}(\mathbf{x},\delta) - \Psi^{i,n}_{\vec{W}}(\mathbf{x},\delta)\right\|$$
$$+ \mathop{\mathbb{E}}_{\vec{W}}\left\|\Psi^{i,n}_{\vec{W}}(\mathbf{x},\delta) - \Psi^{i,n}_{\vec{W}}(\mathbf{x})\right\|$$

Theorem 2.1 now follows from Lemmas 2.11 and 2.12 below, together with Lemma 2.6.

**Lemma 2.11.** *Let $n$ be large enough so that $\epsilon_\sigma(n) \le \frac{1}{2}$ and let $\delta < \frac{\sqrt{\epsilon_\sigma(n)}}{2L+2\lambda(n)}$. Then,*

$$\mathop{\mathbb{E}}_{\vec{W}}\left\|\Psi^i_{\vec{W}}(\mathbf{x},\delta) - \Psi^{i,n}_{\vec{W}}(\mathbf{x},\delta)\right\| \le 12 \cdot (L+1)^2 \cdot (\epsilon_\sigma(n))^{2^{-i}}$$

*Proof.* We will prove the result by induction on $i$. The case $i=0$ is clear as $\Psi^0_{\vec{W}}(\mathbf{x},\delta) = \Psi^{0,n}_{\vec{W}}(\mathbf{x},\delta)$. Fix $i > 0$. For every $\delta < \frac{1}{2}$ and $n$ we have by Lemma 2.10

$$\mathop{\mathbb{E}}_{W^i}\left\|\Psi^i_{\vec{W}}(\mathbf{x},\delta) - \Psi^{i,n}_{\vec{W}}(\mathbf{x},\delta)\right\| \le 2L\delta + \sqrt{2\epsilon_\sigma(n)} + \sqrt{\frac{6L^2}{1-\epsilon_\sigma(n)}\left\|\Psi^{i-1}_{\vec{W}}(\mathbf{x},\delta) - \Psi^{i-1,n}_{\vec{W}}(\mathbf{x},\delta)\right\|} + 2\lambda(n)\delta$$

Taking expectation over $W^1,\dots,W^{i-1}$ we get

$$\mathop{\mathbb{E}}_{\vec{W}}\left\|\Psi^i_{\vec{W}}(\mathbf{x},\delta) - \Psi^{i,n}_{\vec{W}}(\mathbf{x},\delta)\right\|$$

$$\le \quad 2L\delta + \sqrt{2\epsilon_\sigma(n)} + \mathop{\mathbb{E}}_{\vec{W}}\sqrt{\frac{6L^2}{1-\epsilon_\sigma(n)}\left\|\Psi^{i-1}_{\vec{W}}(\mathbf{x},\delta) - \Psi^{i-1,n}_{\vec{W}}(\mathbf{x},\delta)\right\|} + 2\lambda(n)\delta$$

$$\overset{\text{Jensen inequality}}{\le} \quad 2L\delta + \sqrt{2\epsilon_\sigma(n)} + \sqrt{\frac{6L^2}{1-\epsilon_\sigma(n)}\mathop{\mathbb{E}}_{\vec{W}}\left\|\Psi^{i-1}_{\vec{W}}(\mathbf{x},\delta) - \Psi^{i-1,n}_{\vec{W}}(\mathbf{x},\delta)\right\|} + 2\lambda(n)\delta$$

$$\overset{\delta < \frac{\sqrt{\epsilon_\sigma(n)}}{2L+2\lambda(n)}}{\le} \quad 4\sqrt{\epsilon_\sigma(n)} + \sqrt{\frac{6L^2}{1-\epsilon_\sigma(n)}\mathop{\mathbb{E}}_{\vec{W}}\left\|\Psi^{i-1}_{\vec{W}}(\mathbf{x},\delta) - \Psi^{i-1,n}_{\vec{W}}(\mathbf{x},\delta)\right\|}$$

$$\overset{\epsilon_\sigma(n) \le \frac{1}{2}}{\le} \quad 4\sqrt{\epsilon_\sigma(n)} + L\sqrt{12\mathop{\mathbb{E}}_{\vec{W}}\left\|\Psi^{i-1}_{\vec{W}}(\mathbf{x},\delta) - \Psi^{i-1,n}_{\vec{W}}(\mathbf{x},\delta)\right\|}$$

$$\overset{\text{Induction hypothesis}}{\le} \quad 4\sqrt{\epsilon_\sigma(n)} + L\sqrt{12 \cdot 12 \cdot (L+1)^2 \cdot (\epsilon_\sigma(n))^{2^{-i+1}}}$$

$$\le \quad (L+1)\sqrt{12 \cdot 12 \cdot (L+1)^2 \cdot (\epsilon_\sigma(n))^{2^{-i+1}}}$$

$$= \quad 12 \cdot (L+1)^2 \cdot (\epsilon_\sigma(n))^{2^{-i}}$$

$\square$

**Lemma 2.12.** *Fix $i, n, \delta$ and $\epsilon > 0$. There is a constant $D$ such that if $d_1, \dots, d_{i-1} \ge D$ then*

$$\mathop{\mathbb{E}}_{\vec{W}}\left\|\Psi^i_{\vec{W}}(\mathbf{x}) - \Psi^i_{\vec{W}}(\mathbf{x},\delta)\right\| + \mathop{\mathbb{E}}_{\vec{W}}\left\|\Psi^{i,n}_{\vec{W}}(\mathbf{x},\delta) - \Psi^{i,n}_{\vec{W}}(\mathbf{x})\right\| \le \epsilon$$

*Proof sketch (see Appendix A.7 for the formal proof).* Let $B_{i,\delta}$ be the event that for some $j < i$, $|1 - \|\Psi^j_{\vec{W}}(\mathbf{x})\|| > \delta$ or $|1 - \|\Psi^{j,n}_{\vec{W}}(\mathbf{x})\|| > \delta$. We have

$$\mathop{\mathbb{E}}_{\vec{W}}\left\|\Psi^i_{\vec{W}}(\mathbf{x}) - \Psi^i_{\vec{W}}(\mathbf{x},\delta)\right\| = \mathop{\mathbb{E}}_{\vec{W}}\left[\left\|\Psi^i_{\vec{W}}(\mathbf{x})\right\| 1_{B_{i,\delta}}\right] \le \sqrt{\mathop{\mathbb{E}}_{\vec{W}}\left[\left\|\Psi^i_{\vec{W}}(\mathbf{x})\right\|^2\right]}\sqrt{\Pr(B_{i,\delta})}.$$

Similarly,

$$\mathop{\mathbb{E}}_{\vec{W}}\left\|\Psi^{i,n}_{\vec{W}}(\mathbf{x}) - \Psi^{i,n}_{\vec{W}}(\mathbf{x},\delta)\right\| \le \sqrt{\mathop{\mathbb{E}}_{\vec{W}}\left[\left\|\Psi^{i,n}_{\vec{W}}(\mathbf{x})\right\|^2\right]}\sqrt{\Pr(B_{i,\delta})}.$$

Now, the lemma follows by proving that $\mathbb{E}_{\vec{W}}\left\|\Psi^i_{\vec{W}}(\mathbf{x})\right\|^2$ and $\mathbb{E}_{\vec{W}}\left\|\Psi^{i,n}_{\vec{W}}(\mathbf{x})\right\|^2$ are bounded by a constant (independent of $d_0, \dots, d_i$), and that for every $\delta, \epsilon', i$ and $n$, there is a constant $D$ such that if $d_1, \dots, d_{i-1} \ge D$ then $\Pr(B_{i,\delta}) < \epsilon'$. $\square$

# 3 Conclusion and Future work

One of the prominent approaches for explaining the success of neural networks is trying to show that they are capable of learning complex and "deep" models. So far this approach has relatively limited success. Despite that significant progress has been made to show that neural networks can learn shallow models, so far, neural networks were shown to learn only "toy" deep models (e.g. [21, 2, 10, 31]). Not only that, but there are almost no known rich families of deep models that are efficiently learnable by *some* algorithm (not necessarily gradient methods on neural networks). Our paper suggests that random neural networks might be candidate models. To take this approach further, a natural next step, and a central open question that arises from our work, is to show the existence of an algorithm that learns random networks in time that is polynomial both in $\frac{1}{\epsilon}$ and the network size. This question is already open for depth-two ReLU networks with two hidden neurons. We note that as implied by [31], such a result, even for a single neuron, will have to go beyond polynomial approximation of the network, and even more generally, beyond kernel methods.

Our result requires a lower bound $D$ for the network's width, where $D$ is a constant. We conjecture that this requirement can be relaxed, and leave it to future work. Additional open directions are: (i) the analysis of random convolutional networks, (ii) achieving time and sample complexity of $(\bar{d})^{O(\epsilon^{-2})}$ for random networks of any constant depth (and not only for depth two), and (iii) finding a PTAS for random networks of depth $\omega(1)$.

## Acknowledgments and Disclosure of Funding

The research described in this paper was funded by the European Research Council (ERC) under the European Union's Horizon 2022 research and innovation program (grant agreement No. 101041711), and the Israel Science Foundation (grant number 2258/19). This research was done as part of the NSF-Simons Sponsored Collaboration on the Theoretical Foundations of Deep Learning.

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

# A  Missing proofs

## A.1  Some Technical Lemmas

**Lemma A.1.** *If $\sigma$ is L-Lipschitz then $\hat{\sigma}$ is $L^2$-Lipschitz in $[-1, 1]$*

*Proof.* As shown in [14], $(\hat{\sigma})' = \widehat{\sigma'}$. Hence, for $\rho \in [-1, 1]$,

$$
\begin{aligned}
|(\hat{\sigma})'(\rho)| &= \left|\widehat{\sigma'}(\rho)\right| \\
&\leq \|\sigma'\|^2 \\
&\leq L^2
\end{aligned}
$$

$\square$

**Lemma A.2.** $|h_n(x) - h_n(x + y)| \leq 2^n \max(|x|, |x + y|, 1)^n |y|$

*Proof.* It is not hard to verify by induction on Eq. (1) that

$$
|h_n(x)| \leq 2^{n/2} \max(1, |x|^n)
$$

This implies that for $\xi \in [x, x + y]$

$$
\begin{aligned}
|h_n(x) - h_n(x + y)| &= |h_n'(\xi)y| \\
&\overset{Eq.\ (3)}{=} \sqrt{n}|h_{n-1}(\xi)y| \\
&\leq \sqrt{n}2^{n/2} \max(|x|, |x + y|, 1)^n |y| \\
&\leq 2^n \max(|x|, |x + y|, 1)^n |y|
\end{aligned}
$$

$\square$

## A.2  Bounds on $\epsilon_\sigma(n)$

By Eq. (3) if $\sigma$ is differentiable $k$ times then we have $\sigma^{(k)} = \sum_{i=k}^{\infty} \sqrt{\frac{i!}{(i-k)!}} a_i h_{i-k}$. Hence, for $k \leq n + 1$,

$$
\epsilon_\sigma(n) = \sum_{i=n+1}^{\infty} a_i^2 \leq \frac{(n + 1 - k)!}{(n + 1)!} \sum_{i=n+1}^{\infty} \frac{i!}{(i - k)!} a_i^2 \leq \frac{(n + 1 - k)!}{(n + 1)!} \left\|\sigma^{(k)}\right\|^2 \tag{7}
$$

**Lemma A.3.** *For any L-Lipschitz $\sigma$ we have $\epsilon_\sigma(n) \leq \frac{L^2}{n}$.*

*Proof.* By Eq. (7) for $k = 1$ we get

$$
\epsilon_\sigma(n) \leq \frac{1}{n + 1} \|\sigma'\|^2 \leq \frac{L^2}{n + 1}
$$

$\square$

**Lemma A.4.** *For the sigmoid activation $\sigma(x) = \int_0^x e^{-\frac{t^2}{2}} dt$ we have $\epsilon_\sigma(n) \leq 2^{-n}$.*

*Proof.* We have $\sigma^{(k)}(x) = (-1)^{k-1}\sqrt{(k - 1)!}h_{k-1}(x)e^{-\frac{x^2}{2}}$. Indeed, it is not hard to verify it for $k = 1$ and $k = 2$. For $k > 2$ we have via induction that

$$
\begin{aligned}
\sigma^{(k+1)}(x) &= (-1)^{k-1}\sqrt{(k - 1)!} \left[h_{k-1}'(x) - xh_{k-1}(x)\right] e^{-\frac{x^2}{2}} \\
&\overset{Eq.\ (3)}{=} (-1)^k \sqrt{k!}\frac{1}{\sqrt{k}} \left[xh_{k-1}(x) - \sqrt{k - 1}h_{k-2}(x)\right] e^{-\frac{x^2}{2}} \\
&\overset{Eq.\ (1)}{=} (-1)^k \sqrt{k!}h_k(x)e^{-\frac{x^2}{2}}
\end{aligned}
$$

Hence, $|\sigma^{(k)}(x)| \leq |\sqrt{(k - 1)!}h_{k-1}(x)|$, and now Eq. (7) implies that for any $k \leq n + 1$

$$
\epsilon_\sigma(n) \leq \frac{(n + 1 - k)!}{(n + 1)!}(k - 1)! = \frac{(n + 1 - k)!k!}{(n + 1)!k} = \frac{1}{k\binom{n+1}{k}}
$$

Taking $k = \lceil\frac{n+1}{2}\rceil$ we conclude that $\epsilon_\sigma(n) \leq 2^{-n}$.

$\square$

## A.3 Proof of Lemma 2.2

Let $A$ be the event that there is an entry in $\vec{W}$ that is greater than $\sum_{j=0}^{i} d_j$. We have

$$\mathbb{E}_{\vec{W}} \left\| \Phi_{\vec{W}}^{i,n}(\mathbf{x}) - \tilde{\Phi}_{\vec{W}}^{i,n}(\mathbf{x}) \right\| = \mathbb{E} \left[ \left\| \Phi_{\vec{W}}^{i,n}(\mathbf{x}) \right\| \cdot 1_A \right] \leq \sqrt{\mathbb{E} \left\| \Phi_{\vec{W}}^{i,n}(\mathbf{x}) \right\|^2} \sqrt{\Pr(A)}$$

Now, it is not hard to verify that $\mathbb{E} \left\| \Phi_{\vec{W}}^{i,n}(\mathbf{x}) \right\|^2$ is polynomial in $\sum_{j=0}^{i} d_j$ while $\Pr(A)$ converges to $0$ exponentially fast in $\sum_{j=0}^{i} d_j$. Thus, if $\min(d_1, \ldots, d_{i-1})$ is large enough then $\mathbb{E}_{\vec{W}} \left\| \Phi_{\vec{W}}^{i,n}(\mathbf{x}) - \tilde{\Phi}_{\vec{W}}^{i,n}(\mathbf{x}) \right\| < \epsilon$.

## A.4 Proof of Lemma 2.3

We assume that $\tilde{\Phi}_{\vec{W}}^{i,n} = \Phi_{\vec{W}}^{i,n}$, as otherwise $\tilde{\Phi}_{\vec{W}}^{i,n} \equiv 0$, in which case the lemma is clear. Write $\sigma_n(x) = \sum_{k=0}^{n} b_k x^k$ and $h_j(x) = \sum_{k=0}^{j} c_{j,k} x^k$. Via induction on Eq. (1), we have $|c_{j,k}| \leq 2^{\frac{j}{2}}$. Hence,

$$
\begin{aligned}
|b_k| &\leq \frac{1}{\sqrt{\sum_{j=0}^{n} a_j^2}} \sum_{j=0}^{n} |a_j| |c_{j,k}| \\
&\leq \frac{1}{\sqrt{\sum_{j=0}^{n} a_j^2}} \sum_{j=0}^{n} |a_j| 2^{\frac{j}{2}} \\
&\leq \frac{1}{\sqrt{\sum_{j=0}^{n} a_j^2}} \sqrt{\sum_{j=0}^{n} a_j^2} \sqrt{\sum_{j=0}^{n} 2^j} \\
&\leq 2^{\frac{n+1}{2}}
\end{aligned}
$$

Now, let $M_j$ be the maximal sum of coefficients of any polynomial computed by an output neuron of $\Psi_{\vec{W}}^{j,n}$. We next show by induction that $M_j \leq (2\bar{d})^{2 \sum_{k=1}^{j} n^k}$. This will conclude the proof as it will imply that the sum of the coefficients of the polynomial computed by $\Phi_{\vec{W}}^{i,n}$ is at most $(2\bar{d})^2 M_{i-1} \leq (2\bar{d})^{2 \sum_{k=0}^{i-1} n^k} \leq (2\bar{d})^{4n^{i-1}}$. For $j = 0$ we have $M_0 = 1$. For $j \geq 1$ we we have

$$M_j \leq \sum_{k=0}^{n} |b_k| \left( (\bar{d})^2 M_{j-1} \right)^k \leq 2^{\frac{n+1}{2}} \cdot 2 \cdot \left( (\bar{d})^2 M_{j-1} \right)^n \leq \left( (2\bar{d})^2 M_{j-1} \right)^n$$

By the induction hypothesis we have

$$M_j \leq (2\bar{d})^{2n + 2n \sum_{k=1}^{j-1} n^k} = (2\bar{d})^{2 \sum_{k=1}^{j} n^k}$$

## A.5 Proof of Lemma 2.4

We have

$$\mathbb{E}_{W} \| f(W\mathbf{x}) - g(W\mathbf{y}) \| \overset{\text{Jensen Inequality}}{\leq} \sqrt{\mathbb{E}_{W} \| f(W\mathbf{x}) - g(W\mathbf{y}) \|^2}$$

$$= \sqrt{\frac{1}{d_2} \sum_{j=1}^{d_2} \mathbb{E}_{W} (f((W\mathbf{x})_j) - g((W\mathbf{y})_j))^2}$$

Now, the lemma follows from the fact that $\{((W\mathbf{x})_j, (W\mathbf{y})_j)\}_{j=1}^{d_2}$ are independent centered Gaussian vectors with covariance matrix $\begin{pmatrix} \|\mathbf{x}\|^2 & \langle \mathbf{x}, \mathbf{y} \rangle \\ \langle \mathbf{x}, \mathbf{y} \rangle & \|\mathbf{y}\|^2 \end{pmatrix}$.

## A.6 Proof of Lemma 2.8

Let $P_A : \mathbb{R}^d \to A$ a function such that for any $\mathbf{x} \in \mathbb{R}^d$, $\|P_A(\mathbf{x}) - \mathbf{x}\| = d(\mathbf{x}, A)$. Assume w.l.o.g. that $\|\mathbf{x}_1 - P_A(\mathbf{x}_1)\| \leq \|\mathbf{x}_2 - P_A(\mathbf{x}_2)\|$.

**Case I:** $\|\mathbf{x}_2 - P_A(\mathbf{x}_2)\| \leq \|\mathbf{x}_1 - \mathbf{x}_2\|$

Simply define $\tilde{\mathbf{x}}_i = P_A(\mathbf{x}_i)$. We have

$$\|\mathbf{x}_1 - \tilde{\mathbf{x}}_1\| = \|\mathbf{x}_1 - P_A(\mathbf{x}_1)\|, \quad \|\mathbf{x}_2 - \tilde{\mathbf{x}}_2\| = \|\mathbf{x}_2 - P_A(\mathbf{x}_2)\|$$

and

$$\|\tilde{\mathbf{x}}_1 - \tilde{\mathbf{x}}_2\| \leq \|P_A(\mathbf{x}_1) - \mathbf{x}_1\| + \|\mathbf{x}_1 - \mathbf{x}_2\| + \|\mathbf{x}_2 - P_A(\mathbf{x}_2)\| \leq 3\|\mathbf{x}_1 - \mathbf{x}_2\|$$

**Case II:** $\|\mathbf{x}_1 - \mathbf{x}_2\| \leq \|\mathbf{x}_2 - P_A(\mathbf{x}_2)\|$

Define $\tilde{\mathbf{x}}_1 = \tilde{\mathbf{x}}_2 = P_A(\mathbf{x}_1)$. We have

$$\|\mathbf{x}_1 - \tilde{\mathbf{x}}_1\| = \|\mathbf{x}_1 - P_A(\mathbf{x}_1)\|, \quad \|\tilde{\mathbf{x}}_1 - \tilde{\mathbf{x}}_2\| \leq 0\|\mathbf{x}_1 - \mathbf{x}_2\|$$

and

$$\|\mathbf{x}_2 - \tilde{\mathbf{x}}_2\| \leq \|\mathbf{x}_2 - \mathbf{x}_1\| + \|\mathbf{x}_1 - P_A(\mathbf{x}_1)\| \leq 2\|\mathbf{x}_2 - P_A(\mathbf{x}_2)\|$$

## A.7 Proof of Lemma 2.12

Let $B_{i,\delta}$ be the event that for some $j < i$, $|1 - \|\Psi_{\vec{W}}^j(\mathbf{x})\|| > \delta$ or $|1 - \|\Psi_{\vec{W}}^{j,n}(\mathbf{x})\|| > \delta$. We have

$$\mathbb{E}_{\vec{W}} \left\| \Psi_{\vec{W}}^i(\mathbf{x}) - \Psi_{\vec{W}}^i(\mathbf{x}, \delta) \right\| = \mathbb{E}_{\vec{W}} \left[ \left\| \Psi_{\vec{W}}^i(\mathbf{x}) \right\| 1_{B_{i,\delta}} \right] \leq \sqrt{\mathbb{E}_{\vec{W}} \left[ \left\| \Psi_{\vec{W}}^i(\mathbf{x}) \right\|^2 \right]} \sqrt{\Pr(B_{i,\delta})}$$

Similarly,

$$\mathbb{E}_{\vec{W}} \left\| \Psi_{\vec{W}}^{i,n}(\mathbf{x}) - \Psi_{\vec{W}}^{i,n}(\mathbf{x}, \delta) \right\| \leq \sqrt{\mathbb{E}_{\vec{W}} \left[ \left\| \Psi_{\vec{W}}^{i,n}(\mathbf{x}) \right\|^2 \right]} \sqrt{\Pr(B_{i,\delta})}$$

the lemma now follows from the following two claims.

**Claim 1.** $\mathbb{E}_{\vec{W}} \left\| \Psi_{\vec{W}}^i(\mathbf{x}) \right\|^2$ and $\mathbb{E}_{\vec{W}} \left\| \Psi_{\vec{W}}^{i,n}(\mathbf{x}) \right\|^2$ are bounded by a constant (independent of $d_0, \ldots, d_i$).

*Proof.* We have

$$\mathbb{E}_{W^i} \left\| \Psi_{\vec{W}}^i(\mathbf{x}) \right\|^2 = \mathbb{E}_{\mathbf{w}} \sigma^2 \left( \mathbf{w}^\top \Psi_{\vec{W}}^{i-1}(\mathbf{x}) \right)$$

$$\leq 2\sigma^2(0) + 2L^2 \mathbb{E}_{\mathbf{w}} \left( \mathbf{w}^\top \Psi_{\vec{W}}^{i-1}(\mathbf{x}) \right)^2$$

$$= 2\sigma^2(0) + 2L^2 \|\Psi_{\vec{W}}^{i-1}(\mathbf{x})\|^2$$

By induction on $i$, this implies that $\mathbb{E}_{\vec{W}} \left\| \Psi_{\vec{W}}^i(\mathbf{x}) \right\|^2$ is bounded by a constant that depends only on $i$ and $L$ (but not on $d_1, \ldots, d_i$). For $\mathbb{E}_{\vec{W}} \left\| \Psi_{\vec{W}}^{i,n}(\mathbf{x}) \right\|^2$ we have

$$\mathbb{E}_{W^i} \left\| \Psi_{\vec{W}}^{i,n}(\mathbf{x}) \right\|^2 = \mathbb{E}_{\mathbf{w}} \sigma_n^2 \left( \mathbf{w}^\top \Psi_{\vec{W}}^{i-1,n}(\mathbf{x}) \right)$$

Hence, $\mathbb{E}_{W^i} \left\| \Psi_{\vec{W}}^{i,n}(\mathbf{x}) \right\|^2$ is an even polynomial in $\left\| \Psi_{\vec{W}}^{i-1,n}(\mathbf{x}) \right\|$ of degree $\leq 2n$. The polynomial depends only on $\sigma_n$. It therefore enough to show that for any $i$ and $k$, $\mathbb{E}_{\vec{W}} \left\| \Psi_{\vec{W}}^{i,n}(\mathbf{x}) \right\|^{2k}$ is bounded,

by a bound that is independent of $d_0, \ldots, d_i$. We will show that via induction on $i$. For $i = 0$ this is trivial as $\left\| \Psi_{\vec{W}}^{0,n}(\mathbf{x}) \right\|^{2k} \equiv 1$. Fix $i \geq 1$. We have

$$
\underset{W^i}{\mathbb{E}} \left\| \Psi_{\vec{W}}^{i,n}(\mathbf{x}) \right\|^{2k} = \underset{W^i}{\mathbb{E}} \left( \frac{\sum_{j=1}^{d_i} \sigma_n^2 \left( \left( W^i \Psi_{\vec{W}}^{i-1,n}(\mathbf{x}) \right)_j \right)}{d_i} \right)^k
$$

$$
\overset{\text{Jensen inequality}}{\leq} \frac{1}{d_i} \underset{W^i}{\mathbb{E}} \sum_{j=1}^{d_i} \sigma_n^{2k} \left( \left( W^i \Psi_{\vec{W}}^{i-1,n}(\mathbf{x}) \right)_j \right)
$$

$$
= \underset{\mathbf{w}}{\mathbb{E}} \, \sigma_n^{2k} \left( \mathbf{w}^\top \Psi_{\vec{W}}^{i-1,n}(\mathbf{x}) \right)
$$

The last expression is an even polynomial in $\| \Psi_{\vec{W}}^{i-1,n}(\mathbf{x}) \|$. The polynomial depends only on $2k$ and $n$. By the induction hypothesis we conclude that $\mathbb{E}_{\vec{W}} \left\| \Psi_{\vec{W}}^{i,n}(\mathbf{x}) \right\|^{2k}$ is bounded by a bound that is independent from $d_0, \ldots, d_i$. $\qquad \square$

**Claim 2.** *For every $\delta, \epsilon'$, $i$ and $n$, there is a constant $D$ such that if $d_1, \ldots, d_{i-1} \geq D$ then $\Pr(B_{i,\delta}) < \epsilon'$.*

*Proof.* We will prove the lemma by induction on $i$. For $i = 1$ this is immediate as $\Pr(B_{i,\delta}) = 0$. Fix $i \geq 2$. Let $\delta'$ be small enough so that if $|\|\mathbf{x}\| - 1| \leq \delta'$ then

$$
\left| \underset{\mathbf{w}}{\mathbb{E}} \sigma^2(\mathbf{w}^\top \mathbf{x}) - 1 \right| < \frac{\delta}{4} \text{ and } \left| \underset{\mathbf{w}}{\mathbb{E}} \sigma_n^2(\mathbf{w}^\top \mathbf{x}) - 1 \right| < \frac{\delta}{4}
$$

and

$$
\left| \underset{\mathbf{w}}{\mathbb{E}} \sigma^4(\mathbf{w}^\top \mathbf{x}) - \underset{X}{\mathbb{E}} \sigma^4(X) \right| < 1 \text{ and } \left| \underset{\mathbf{w}}{\mathbb{E}} \sigma_n^4(\mathbf{w}^\top \mathbf{x}) - \underset{X}{\mathbb{E}} \sigma_n^4(X) \right| < 1
$$

we have

$$
\Pr(B_{i,\delta}) \leq \Pr(B_{i,\delta} | B_{i-1,\delta'}^c) + \Pr(B_{i-1,\delta'})
$$

By Chebyshev inequality, $\Pr(B_{i,\delta} | B_{i-1,\delta'}^c) < \frac{\epsilon'}{2}$ for sufficiently large $d_{i-1}$. By the induction hypothesis, $\Pr(B_{i-1,\delta'}) < \frac{\epsilon'}{2}$ for sufficiently large $d_1, \ldots, d_{i-2}$ $\qquad \square$

