# OpenReview forum: "Most Neural Networks Are Almost Learnable"
_NeurIPS.cc/2023/Conference — NeurIPS 2023 poster_

### Official Review · Reviewer_cQDt · 2023-06-21

**Soundness:** 3 good
**Presentation:** 2 fair
**Contribution:** 3 good
**Rating:** 7
**Confidence:** 3

**Summary:**

The paper examines the extent to which neural networks learn deep models. Given the fact that there almost do not exist any 'rich' families of constant depth random neural networks that are efficiently learnable by some algorithm, the authors assert the existence of an algorithm that approximates these families of networks that are comprised by Lipschitz activation functions. To achieve this, they substitute each activation function with a low degree polynomial approximation (Hermite polynomial), and then show that the approximation is good enough, allowing for a standard SGD on the neural networks to be a PTAS.

**Strengths:**

-The paper examines the expressivity and learnabilty of random neural networks. This is a fundamental problem in the Theory of Deep Learning and the authors manage to address it by bounding the expected learning time, i.e. the # iterations for determining the weights of the networks approximately (if I undestand correctly Th.1.1.). This allows for a possible impact of the paper in this field.

-From a first reading the results seem new.

**Weaknesses:**

-The presentation could be improved. The mathematical formulation could be more explained, e.g. with an example of a toy neural network, where the weights and random initialization can be displayed pictorially.

-There could be a clear comparison to the previous approximation attempts (of the literature) to show the novelty in the current PTAS. E.g. in the 1.2. section there is a mention in Chen et al. [7] who provided an algorithm for learning deeper ReLU networks, with complexity polynomial in the input dimension but exponential in the other parameters. The current approach of this paper (from what I understand) considers fixed depth networks and asserts the existence of a polynomial which can be 'close' (approximately) to the learning outcome of the trained neural network: after i iterations, it can approximate it expectedly in 'well' (ideally constantly, as the iterations 'i' tend to infinity). This can be contrasted to previous approaches mentioned in 1.2 about learning depth-3 networks, hardness result of the setting where the target
network is random and an adversary chooses some input distribution which needs to be learned by the random target network etc.

**Questions:**

Please refer to the limitations below.

**Limitations:**

-The paper does not attempt to make an experiment as a proof of concept. For example it could try several neural networks of constant depth and Lipschitz activations and vary the rest of the hyperparameters randomly. Then there could be some kind of benchmarking across several  neural networks that measures how good the approximation is, while the learning runs in polynomial time. The running time is also interesting in terms of such reporting.
Although the paper is theoretical, it discusses a potential application of such neural networks (their polynomial approximation). This is the reason why it would be interesting to offer a numerical result of the efficiency, accuracy and complexity of learning a polynomial.

-It seems to be inherently difficult to find the polynomial of Th. 1.1 in order to prove the above point. Is this within the scope of this paper, besides the existence result?

---

> ### Author Rebuttal · Authors · 2023-08-09
>
> We thank the reviewer for the positive review. We will improve the writing according to the outlined suggestions.
>
> We next address the reviewer’s concerns:
>
> - Experiments: Thanks for the suggestion. We will run experiments and consider whether to add them to the final version.
>
> - Finding the polynomial: It is indeed not clear how to find the polynomial of the theorem. Yet, for the algorithms of the paper, existence is enough.

---

> > ### Comment · Reviewer_cQDt · 2023-08-14
> > **Answer to authors during the rebuttal**
> >
> > I thank the authors for their response. I am inclined to keep my score for acceptance for now.

---

### Official Review · Reviewer_PwvA · 2023-06-22

**Soundness:** 3 good
**Presentation:** 2 fair
**Contribution:** 3 good
**Rating:** 4
**Confidence:** 3

**Summary:**

The paper proves that a sufficiently wide neural network which is randomly sampled according to Xavier initialization can be learned in polynomial time (in the network size) up to an additive error $\epsilon$ if the numbers of layers, the Lipschitz-constant of the activation function, and $\epsilon$ are fixed constants.

**Strengths:**

- According to the authors, this is the first positive learnability result of this type for arbitrary deep neural networks. This seems to be a major milestone in learning theory for deep networks.
- The obtained learning algorithm is actually SGD on ReLU networks, so exactly what people use in practice.

**Weaknesses:**

- The paper is crazily technical already in the main part, not even taking into account the appendix. Within the limited time available for a NeurIPS review, it is impossible to verify mathematical correctness. I tend to believe that this is inherent to the result and not the authors' fault, but I wonder if NeurIPS is actually the right venue to publish such a result. Maybe the authors should submit to a journal instead, where reviewers have enough time to verify mathematical correctness. If submitting to NeurIPS, I think the emphasis should be much more on providing intuitions of the proofs than on technical details.
- The result is only valid if the network is sufficiently wide enough. Looking through the proof it seems that the required depth $D$ depends with a double-factorial on $n$, which in turn depends on the desired precision $\epsilon$. This will be astronomically large for any reasonable value of $\epsilon$. I do not think that any practical neural network can be so wide.
- Obviously also all other constants seem to be huge / high exponential dependicies on the fixed quantities. But I think this is fine for a theoretical result, it does not concern me as much as the lower bound on the width mentioned above.

Minor comments:

- line 26: I think it would be good to define what you mean by "ReLU-like" already here.
- line 39: you should say that p is a polynomial.
- line 47: it is confusing to call the neuron n, because n denotes the degree of the polynomials to approximate sigma.
- line 60: I find this sentence grammatically weird "Thus, those PTAS can be standard SGD on neural networks". What do you mean here?
- line 63: erf not defined.
- line 65: quasi-polynomial refers to the dependence on epsilon and \bar{d}, still assuming constant i. Point this out.
- line 144: what are the a_i in the definition of \hat(\sigma)? These are only defined a few lines later, please fix this.
- line 196: soften -> softened

**Questions:**

- The result seems to assume realizability of the data, that is, it assumes that the ground truth is actually represented by a Xavier-initialized neural network. Did I understand this correctly and can you say anything about the non-realizable setting (where the data can be anything, but you compare yourself only against the error of a best-possible neural network)?
- How much of a restriction is it to restrict the input data to the boundary of the sphere?
- For sigmoid and "ReLU-like" activation functions, you prove a better dependence on $\epsilon$. Can you obtain something similar for the actual ReLU activation function? If not, what are the obstacles?

**Limitations:**

Generally the authors state all mathematical assumptions in a proper way. However, I think two limitations should be discussed earlier / in more detail:

- The issue with the minimum width $D$ (see weaknesses) should be mentioned in the abstract already and pointed out during the introduction. It seems to be an actual important restriction of the validity of the result.

- There is an issue with the sequence of drawing the random objects of the instance, as the authors point out themselves in lines 123 - 125. As previous work shows, the result can only hold with this particular sequence. While this is of course implicitly captured by the sequence of expectations in Thm. 1.1, I think the authors should point this out when they state Thm. 1.1 already, and not only hidden in the related work part.

---

> ### Author Rebuttal · Authors · 2023-08-09
>
> We thank the reviewer for the detailed review. We will improve the writing according to the outlined suggestions.
>
> First, the reviewer says that our paper “seems to be a major milestone in learning theory for deep networks” and does not raise soundness concerns. We believe that this may qualify the paper for a rating better than 4.
>
> We next address the reviewer’s concerns and questions:
>
> - Dependence on constants: The constant $D$ is of the form $L^{n^i}$ where $L$ is the Lipschitz constant of the activation function. We conjecture that the theorem is correct with respect to any constant. We will clarify it in the final version.
>
> - Non-Realizable setting: You understood correctly. If $1-\epsilon_0$ fraction of the distribution is realizable by a random network, then our bounds are valid, up to an additive error of $\epsilon_0$.
>
> - *“How much of a restriction is it to restrict the input data to the boundary of the sphere?”:*
> We can relax this requirement and assume that the inputs are close to the sphere. This will result in an additional factor in the bounds.
>
> - ReLU-like vs. ReLU: As opposed to ReLU, ReLU-like activations have good polynomial approximations. Hence, our results give better bounds for these functions.

---

> > ### Comment · Reviewer_PwvA · 2023-08-13
> >
> > I would like to thank the authors for their response to my review and for answering my questions. While I agree with the authors that the result could qualify for a better rating than 4, I remain hesitant to recommend acceptance due to the lack of intuitive explanations for the very technical content in the paper. A paper at NeurIPS should be comprehensible for the very diverse ML community.

---

> > > ### Author Response · Authors · 2023-08-13
> > >
> > > We will add more intuitive explanations to make the paper easier to read for a more diverse audience. We note that the proof approach is simple: for a random network, we can approximate each activation sufficiently well with low-degree Hermite polynomials. The calculations required to prove our result are indeed non-trivial, but we believe that there are many papers in NeurIPS which are much more technical than ours. The mathematical tools required to understand our proof are approachable to a wide audience.

---

> > > > ### Comment · Reviewer_PwvA · 2023-08-14
> > > >
> > > > I thank the authors for sharing their perspective. Personally, I see limited value in a paper if more than half of the main part consists of technical calculations that are hard to read and hard to verify, even if the result is a good contribution. I acknowledge that this is a subjective impression. I encourage the authors to keep their promise to focus more on intuitions in the next version of the paper.

---

> > > > > ### Comment · Reviewer_PwvA · 2023-08-14
> > > > >
> > > > > Additionally, I still see it as a major drawback of the result that it is only valid if the width of the network is huge. As I understand the rebuttal, the authors conjecture that this assumption is not necessary. Maybe the authors can comment more on what's the intuitive reason why their proof needs this assumption and what obstacles one needs to overcome to remove it.

---

> > > > > > ### Author Response · Authors · 2023-08-14
> > > > > >
> > > > > > We use the lower bound on the width in order to show that w.h.p. the norms of all layers are approximately $1$. This is done with a concentration-of-measure argument that requires the lower bound. When the scales of the norms are approximately $1$, our approximation with Hermite polynomials performs well. We hope that this assumption can be avoided by using a different polynomial approximation of the activations, which is more resistant to a variation in the scales.

---

> > > > > > > ### Comment · Reviewer_PwvA · 2023-08-14
> > > > > > >
> > > > > > > I thank the authors for the explanation. To me, removing this assumption would make the result much more compelling.

---

### Official Review · Reviewer_JZMD · 2023-07-06

**Soundness:** 3 good
**Presentation:** 2 fair
**Contribution:** 3 good
**Rating:** 6
**Confidence:** 4

**Summary:**

The work shows that there is a PTAS for learning a random xavier feedforward network. They show rigorously that the algorithm runs in time and sample complexity $d^{t}$ where $t = poly(\frac{1}{\varepsilon})$. This is quite an impressive result since it is a distribution-free result in terms of the input $x$. Furthermore, a corollary of this result is that it shows that SGD can learn these networks.

**Strengths:**

1. This is a technically challenging and impressive result.
2. Typically, learnability results for Neural Nets are restricted to 2-layer shallow networks. To extend the result to constant depth networks is quite impressive even if they are only random.

**Weaknesses:**

1. The paper can be better written. While the technical content is good, I feel there could be more intuition and better presentation. Proofs could be moved the appendix and detailed proof sketches could be included.
2. Since the PTAS is for learning random networks, it's practical value is questionable.
3. I found certain aspects of the theory hard to understand - I have included them in the review as questions.

**Questions:**

1. The abstract mentions Xavier networks - shouldn't this only be feedforward networks? Or is it easy to extend to other kinds?
2. I understand that $\epsilon_{\sigma}(n)$ refers to the error when approximating the activation by a polynomial of degree $n$. I would like to see a few lines explaining the intuition behind this quantity and why it is important.
3. The input space is restricted to be **on** the unit sphere. Can this be relaxed to handle $\lVert x \rVert \leq 1$?
4. Theorem 1.1 is stated as "For any weights $W$", but the bound takes an expectation over $W$. Is it on average or "for any"?
5. The notation can be cleaned up. For example in Line 66, consider saying $d^t$ where $t = \mathcal{O}(dots)$
6. Why does Corollary 1.2 consider $\epsilon, i$ as constants? Isn't it just for any $\epsilon, i$? Is it possible to indicate the dependence on $i$ in the bound?
7. Is the claim of Corollary 1.2 that SGD can be used to approximate a Xavier network starting from a Xavier network on average? I understand that this is non-trivial, but I am not able to get a good intuition as to why this is important.
8. I feel line 113-116 provides excellent motivation for the work. I would recommend the authors to highlight it in the introduction.
9. $\delta_{ij}$ is the dirac-delta function?
10. In Lemma 2.7, it says fix $W$ and then takes $\mathbb{E}_W$. I don't understand this.
11. Is the application of Jensen's Inequality in line 233 obvious? $\psi$ isn't convex as a function of all the $W$'s right? Or is the expectation only with respect to the last layer's weights?
12. I think the work is good, but presentation can be improved. I would recommend moving the proofs to the appendix and highlighting intuition and explaining the result regarding SGD in more detail.

**Limitations:**

I feel the authors should include a more detailed limitations section where they address some of the concerns I have listed above as questions (if relevant).

---

> ### Author Rebuttal · Authors · 2023-08-09
>
> We thank the reviewer for the detailed review. We will improve the writing according to the outlined suggestions.
>
> We next address the reviewer’s questions:
>
> - *“The abstract mentions Xavier networks - shouldn't this only be feedforward networks? Or is it easy to extend to other kinds?”:*
> Our techniques are not limited to feedforward networks, and we are certain that they can be used to derive similar results for more architectures. This is mostly left for future work.
>
> - *“I understand that $\epsilon_\sigma(n)$ refers to the error when approximating the activation by a polynomial of degree  $n$. I would like to see a few lines explaining the intuition behind this quantity and why it is important”:*
> In the proof we essentially replace the activation by its polynomial approximation. Thus, it is natural to expect that our bound will depend on the extent to which the activation can be approximated by a polynomial. We will add an explanation to the final version.
>
> - *“The input space is restricted to be on the unit sphere. Can this be relaxed to handle $\lVert x \rVert \le 1$?”:*
> We can relax this requirement and assume that the inputs are close to the unit sphere. This will result in an additional factor in the bounds.
>
> - *“Theorem 1.1 is stated as ‘For any weights $W$’, but the bound takes an expectation over
> $W$. Is it on average or ‘for any’?”:*
> The polynomial $p_W$ is defined for any $W$, and the bound is valid in expectation.
>
> - *“Why does Corollary 1.2 consider $\epsilon, i$ as constants? Isn't it just for any $\epsilon, i$? Is it possible to indicate the dependence on $i$ in the bound?”:*
> The dependence on the constant can be derived from Theorem 1.1. We write the corollary to emphasize that our main theorem implies a PTAS.
>
> - *“Is the claim of Corollary 1.2 that SGD can be used to approximate a Xavier network starting from a Xavier network on average? I understand that this is non-trivial, but I am not able to get a good intuition as to why this is important”:*
> The importance is that this implies that standard training algorithms can learn most neural networks. Thus, this can be seen as a partial explanation for the success of NN, which is a central open question these days.
>
> - *“I feel line 113-116 provides excellent motivation for the work. I would recommend the authors to highlight it in the introduction”:*
> Will do. Thanks!
>
> - *“$\delta_{ij}$ is the dirac-delta function?”:*
> It is the Kronecker delta. We will clarify this.
>
> - *“In Lemma 2.7, it says fix $W$ and then takes $E_W$. I don't understand this”:*
> It should be replaced with “let $W$ be a Xavier matrix”.
>
> - *“Is the application of Jensen's Inequality in line 233 obvious?”:*
> It follows from the concavity of the square root function.

---

> > ### Comment · Reviewer_JZMD · 2023-08-12
> > **Response to the Authors**
> >
> > I thank the reviewers for their response. Since the results on extending it to more architectures is left as future work, I would recommend the authors specify that the results as they are proven are only for feedforward networks. It can be left as a remark that they believe these results can be extended. Other than that, I am satisfied by the reviewer's responses and am increasing my score.

---

> > > ### Author Response · Authors · 2023-08-12
> > >
> > > Thanks. We will add such a remark.

---

### Official Review · Reviewer_mRGU · 2023-07-08

**Soundness:** 4 excellent
**Presentation:** 4 excellent
**Contribution:** 3 good
**Rating:** 7
**Confidence:** 3

**Summary:**

This work presents a polynomial-time approximation scheme (PTAS) for learning random Xavier networks of depth $i$ up to a fixed additive error of $\epsilon$ with respect to any distribution on the hypersphere. For a fixed $\epsilon$, the time and sample complexity is polynomial w.r.t. $\bar{d}$, the total parameter size of the neural network. The work also provides improved bounds for $\bar{d}^{polylog(\epsilon^{-1})}$ for special activation functions such as sigmoid.

Prior work has several limitations, mainly that most consider only one hidden-layer-layer networks. The most comparable work is Chen et al. [7] which provides an algorithm for learning deeper than one layer ReLU networks, yet while the complexity is polynomial in $\bar{d}$, it is exponential in paramaters such as the number of hidden units, depth, spectral norm of the weight matrices, and Lipschitz constant. In fact, there are a plethora of negative results regarding hardness of learning. This work avoids these hardness results by fixing an input distribution and then drawing a random weights of the network.

The main idea of the paper is to approximate the activation function sufficiently close to the original by replacing it with a polynomial approximation using Hermite polynomials (Theorem 1.1).  The paper calls this a "shadow network" and requires that the shadow network's coefficients are polynomially bounded. Given the extensive work on learning polynomials efficiently, the authors can leverage these algorithms, such as SGD on learning polynomial approximations, to obtain a PTAS for learning random networks to a sufficiently small error.

**Strengths:**

- This work studies an important problem of learning deeper-than-two-layers neural networks efficiently. Given the high theoretical relevance of the problem regarding the effectiveness of deep learning, the significance of studying such problem is justified.
- I find the presentation of the paper quite clear, though it may be helpful in describing prior work and results on learning polynomial approximations using Hermite polynomials. Otherwise, the related work and comparison is to prior work are clear.
- The main result provides a PTAS that is distribution-free in its input distribution. Prior work did not get such flexible results and this work achieves this in part by incorporating a common initiailization method (random Xavier networks) and studying "average-case" results instead of worst-case instances.



**Weaknesses:**

I do not see any notable weaknesses. I have some questions, which may be seen as weaknesses, that I will list below.

**Questions:**

- Regarding constant $D(n,i,\sigma)$: It is not entirely clear to me what $D$ explicitly is with respect to the parameters of $n,i,\sigma$. Since $D$ is a lower bound to the number of neurons in each layer, this quantity seems highly relevant and would be helpful to make explicit in the main explicit theorem (like Theorem 2.1). My question is the following: how does the lower bound $D$ compare to prior work?  How would the results change if you do not have an explicit constant $D$ and assume $D := \min(d_1, ..., d_{i-1})$? Would such guarantee exist given the current techniques and what would the guarantees be w.r.t. this new $D$?
- L140: When defining Hermite polynomials, is $\delta_{ij}$ defined? I think this is missing and should be added to Section 2.1.
- L144: When defining dual activation, $a_i$ is undefined yet. Maybe reordering the definition so that dual activation comes after defining $\sigma$ with Hermite polynomials would be clearer.

**Limitations:**

See Questions. Other than that, the authors are clear with the limitations and comparison to prior work.

---

> ### Author Rebuttal · Authors · 2023-08-09
>
> We thank the reviewer for the positive review.
>
> Regarding the questions:
> 1. This is a good question. In our result, we need a lower bound $D$ on the width which is of the form $L^{n^i}$, where $L$ is the Lipschitz constant of the activation function. We also conjecture that the theorem is correct with respect to any constant. Prior work considered significantly different settings, but a lower bound on the width is usually not required. We will discuss it in the final version.
> 2. Right. We will add it.
> 3. We will reorder.

---

> > ### Comment · Reviewer_mRGU · 2023-08-14
> > **Reply to Authors**
> >
> > My questions are answered. Thank you for the response.

---

### Decision · Program_Chairs · 2023-09-21

**Decision:**

Accept (poster)

**Comment:**

The reviewers have generally agreed that this paper presents a significant contribution to the field of deep learning theory. The paper's main strength lies in its novel approach to learning random Xavier networks of depth, providing a polynomial-time approximation scheme (PTAS) that is distribution-free in its input distribution. This is a significant advancement over previous work, which has been largely limited to one hidden-layer networks.

The reviewers have raised some concerns regarding the presentation and clarity of the paper, suggesting that the authors could provide more intuition behind their results and improve the explanation of their mathematical formulations. However, these concerns do not detract from the overall quality and impact of the paper.

Some reviewers have also suggested that the authors could provide a numerical proof of concept or discuss the practical implications of their results in more detail. While these suggestions could potentially enhance the paper, they are not essential for the acceptance of the paper.

In light of the above, the paper is accepted for publication. The authors are encouraged to take the reviewers' feedback into account when preparing the final version of their paper.